**communications** engineering

# Beyond 350 GHz: Single-channel 112 Gbps photonic wireless transmission at 560 GHz using soliton microcombs
Yu Tokizane [1,10,11] ✉, Hiroki Kishikawa[1,10,11], Takumi Kikuhara[2], Miezel Talara[1], Yoshihiro Makimoto[1,3], Kodai Yamaji[2], Yasuhiro Okamura[1,4], Kenji Nishimoto[1], Eiji Hase [1,10], Isao Morohashi[5], Atsushi Kanno[5,6], Shintaro Hisatake[7], Naoya Kuse [1,8], Tadao Nagatsuma [1,9] & Takeshi Yasui [1,8] ✉

Sixth-generation (6 G) back-haul links will require terahertz (THz) carriers above 350 GHz to escape the congested 300 GHz band and support >100 Gbps data rates. Photonic THz transmitters have so far remained below 350 GHz because high-frequency photomixing suffers from phase noise and power limits. Here we demonstrate single-channel wireless transmission at 560 GHz using a fibre-packaged silicon-nitride soliton microcomb as a compact, low-phase-noise optical reference. A high numerical aperture, UV-bonded fibre interface sustains soliton operation for more than 24 hours with 1 W pump power. We phase-lock two distributed-feedback lasers (DFBs) to adjacent comb lines and photomix them in a uni-traveling-carrier photodiode, generating a 560 GHz carrier that bears in-phase and quadrature modulation. We achieve hard-decision forward-error-correction-qualified quadrature phase-shift keying and 16-quadrature amplitude modulation (16QAM) transmissions at 42 and 28 GBaud, respectively, attaining a record 112 Gbps data rate at 560 GHz. Relative to free-running DFBs, microcomb-locked photomixing cuts carrier linewidth and improves 16QAM error-vector magnitude. The results establish soliton microcombs as compact and scalable frequency references for >100 Gbps sub-THz links and chart a path toward compact 6 G back-haul radios.

The surge in mobile data traffic has exposed the bandwidth ceiling of fifth-generation (5 G) networks, which are tied to sub-28 GHz millimetre-wave bands. Meeting sixth-generation (6 G) targets therefore requires tapping the 0.3–1 THz window[1,2], whose ≥100 GHz slices can sustain ultra-high-speed mobile links[3–5]. Yet free-space loss scales with the square of frequency, forcing ultra-dense small cells and airborne relay nodes, and shifting the back-haul burden from fibre to high-capacity wireless channels[6]. While spectrum near 300 GHz is already allocated for access and sensing applications[7], frequency bands above 350 GHz remain unallocated and offer contiguous bandwidths, making them attractive for replacing fibre-based backhaul with next-generation high-capacity wireless links. Realizing wireless transmission in these higher-frequency bands requires the development of high-performance THz transmitters capable of broadband operation above 350 GHz.

Conventional electronic THz transmitters, based on frequency multiplication, are effective up to 300 GHz but face intrinsic limitations in scalability, phase noise, and power efficiency at higher frequencies. As a result, electronic THz transmission systems are effectively limited to frequencies below 350 GHz, whereas photonic THz transmitters have emerged as promising candidates that overcome the 350 GHz barrier and offer scalability towards higher-frequency bands[8]. Among these, photomixing, a photonic THz generation technique[5] that combines optical frequency beating with a uni-traveling-carrier photodiode (UTC-PD)[9,10], offers broadband scalability and inherently low phase noise, making it a strong alternative for THz transmission[11–13]. However, conventional photomixing employing dual-wavelength near-infrared continuous-wave lasers requires bulky laser systems and suffers from high phase noise due to instability

[1]Institute of Post-LED Photonics (pLED), Tokushima University, Tokushima, Japan. [2]Graduate School of Sciences and Technology for Innovation, Tokushima University, Tokushima, Japan. [3]Tokushima Prefectural Industrial Technology Center, Tokushima, Japan. [4]Center for Higher Education and Digital Transformation, University of Yamanashi, Kofu, Japan. [5]National Institute of Information and Communications Technology, Koganei, Japan. [6]Department of Electrical and Mechanical Engineering, Nagoya Institute of Technology, Nagoya, Japan. [7]Department of Electrical, Electronic and Computer Engineering, Gifu University, Gifu, Japan. [8]Institute of Photonics and Human Health Frontier (IPHF), Tokushima University, Tokushima, Japan. [9]Institute for Photon Science and Technology, The University of Tokyo, Tokyo, Japan. [10]Present address: Institute of Photonics and Human Health Frontier (IPHF), Tokushima University, Tokushima, Japan. [11]These authors contributed equally: Yu Tokizane, Hiroki Kishikawa. ✉e-mail: tokizane@tokushima-u.ac.jp; yasui.takeshi@tokushima-u.ac.jp

between the two lasers. To overcome these limitations, phase-stabilized optical frequency combs (OFCs)[14] or stimulated Brillouin scattering lasers[15] have been introduced as frequency references; however, their system complexity and physical bulk hinder practical implementation. Recently, Kerr microresonator-based OFCs (microcombs)[16–18] have emerged as compact, CMOS-compatible, low-phase-noise optical sources suitable for photonic THz generation. Microcombs produce multiple phase-coherent spectral lines with user-configurable spacing ($f_{rep}$), enabling direct photo-mixing of adjacent comb lines for low phase noise THz generation without optical frequency multiplication[19–23]. Owing to these advantages, micro-combs have been adopted in recent experimental demonstrations of THz wireless transmission.

To date, microcomb-based THz transmission has achieved soft-decision forward error correction (SD-FEC)-qualified peak data rates of 250 Gbps[24] and 240 Gbps[22]. However, when evaluated under the more stringent hard-decision FEC (HD-FEC) threshold, these correspond to effective throughputs of 200 Gbps with self-injection-locked soliton microcombs (50 GHz repetition rate; 50-Gbaud 16-quadrature amplitude modulation (16QAM) at a 300 GHz carrier)[24] and 160 Gbps with Kerr microcombs generated in fibre Fabry-Pérot resonators (FFPRs, 283 GHz repetition rate; 40-Gbaud 16QAM at a 283 GHz carrier)[22]. Nevertheless, such demonstrations remain confined to the 300 GHz band, where device technologies are more mature and atmospheric attenuation is moderate. In contrast, pho-tonic THz transmission beyond 350 GHz remains largely unexplored, limited by challenges in carrier generation, signal stability, and system integration. In the 560 GHz band, prior demonstrations have been limited to low-speed transmission: soliton microcomb photomixing combined with Schottky barrier diode detection achieved 2 Gbps on-off keying (OOK) transmission[25], and subsequent improvements employing optical injection locking (OIL) of distributed feedback (DFB) lasers and double-sideband modulation supported binary phase-shift keying (BPSK) and quadrature phase-shift keying (QPSK) formats[26]. In addition, these demonstrations face practical limitations associated with soliton microcombs, including bulky free-space coupling, thermal drift-induced alignment degradation, and instability under high-power pumping due to thermal wobbling of lensed fibres. These issues restrict stable soliton operation to short durations, posing a barrier to continuous THz generation and reliable long-term transmission.

To overcome these limitations and advance microcomb-based THz transmission beyond 350 GHz, we present a scalable photonic transmitter-electronic receiver architecture featuring two key innovations. First, we implement in-phase and quadrature (IQ) modulation combined with sub-harmonic mixing (SHM), enabling high-order, broadband signal genera-tion and high-sensitivity heterodyne detection. Second, we introduce a robust fibre-coupled microcomb packaging scheme that enhances mechanical compactness, thermal robustness, and high-power pumping capability, overcoming free-space coupling constraints. These advances enable long-term stable soliton operation and facilitate continuous high-speed wireless transmission at HD-FEC-qualified rates up to 112 Gbps, including 42 GBaud QPSK and 28 GBaud 16QAM formats at a 560 GHz carrier. Our results represent a practical pathway toward compact, low-phase-noise, photonic THz transmitters capable of supporting next-generation mobile backhaul and photonic-enabled 6 G networks operat-ing above 350 GHz.

## Results
### Direct fibre-coupled soliton microcomb with 560 GHz line spacing
**Interface.** Figure 1 outlines the compact direct fibre-to-chip interface that excites and collects light from a silicon-nitride (SiN) ring micro-resonator (custom, LIGENTEC, S.A., free spectral range (FSR) = 560 GHz, Q ≈ 1 × 10⁶). A standard single-mode fibre (SMF) is fusion-spliced to a high numerical aperture (high-NA) SMF (mode field diameter (MFD) = 3.2 μm) and permanently bonded to the access waveguide via a single-core fibre array and UV-curable adhesive (Fig. 2a). The adhesive

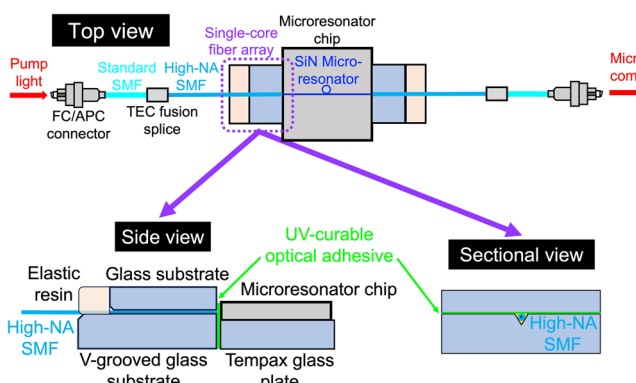

Fig. 1 | Direct fibre-coupled interface for 560 GHz soliton microcomb genera-tion. A compact, robust fibre-to-chip coupling configuration for a silicon nitride (SiN) ring microresonator (free spectral range (FSR) = 560 GHz, Q ≈ 1 × 10⁶). A high-numerical-aperture (NA) single-mode fibre (SMF), fusion-spliced to a stan-dard SMF, is directly bonded to the microresonator's input waveguide using a single-core fibre array and UV-curable adhesive. The top, side, and sectional views illustrate the use of a V-grooved glass substrate and Tempax glass plate for mechanical reinforcement and precise alignment. An identical assembly is applied at the output port. This packaging approach replaces conventional free-space coupling, sub-stantially reducing the optical setup footprint and enhancing thermal stability for sustained soliton microcomb operation, and enabling high-power optical excitation exceeding 1 W.

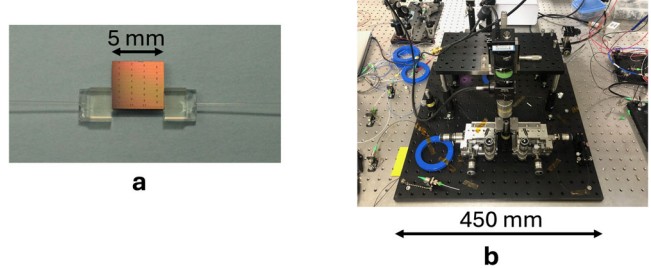

Fig. 2 | Direct fibre-to-chip coupling versus conventional free-space coupling. a Photograph of the direct fibre-to-chip coupling configuration developed in this study, showing a silicon nitride microresonator chip permanently bonded to a SMF via a single-core fibre array (footprint = 5 mm).b Photograph of a conventional free-space coupling setup using a lensed SMF to couple pump light into and out of the microresonator (footprint = 450 mm). The direct interface reduces system size by almost two orders of magnitude, enhancing mechanical robustness and thermal stability.

not only provides mechanical stability but also helps to suppress Fresnel losses by reducing refractive-index mismatch at the interface. The same assembly is employed at the output port, resulting in a footprint sub-stantially smaller than that of conventional lensed-fibre coupling[27] (Fig. 2b).

**Soliton microcomb generation.** Soliton microcomb was generated by injecting a carrier-suppressed single-sideband (CS-SSB) pump light into the fibre-coupled SiN resonator (Fig. 3). Pump light from a 1559.75 nm external-cavity laser diode (linewidth = 500 kHz, power = 20 mW) was fed to a dual-parallel Mach-Zehnder modulator (DP-MZM) to generate a CS-SSB tone (power ≈ 1.4 mW). The modulation frequency of the DP-MZM was linearly swept using a voltage-controlled oscillator, resulting in an effective optical frequency sweep rate of approximately 100 PHz/s, which enabled rapid traversal of the cavity resonance and reliable initiation of soliton formation. After erbium-doped fibre amplification to 1 W and polarisation optimisation, the pump light was launched through the high-NA SMF interface into the microresonator. By sweeping the

effective pump frequency from the short-wavelength side to the long-wavelength side of the cavity resonance, dissipative Kerr-soliton formation was deterministically initiated[28,29]; residual pump was removed with an optical band-stop filter (OBSF). In the present experiment, a pump-to-comb conversion efficiency of ~0.1% (~1 mW comb power for 1 W pump) was obtained, which is typical for single-ring SiN soliton microcombs optimized for robust and deterministic soliton formation rather than power extraction. Importantly, recent studies have shown that soliton microcomb efficiency can be dramatically enhanced, up to 30-50%, by employing coupled-resonator architectures, even with moderate Q factors on the order of $10^6$, as reported previously[30,31], indicating that the relatively low efficiency observed here is not a fundamental limitation but a design choice prioritizing stability and robustness. An otherwise identical bench-top setup using a lensed-fibre coupler (Fig. 2b) served as a free-space reference[27]. Full experimental details are provided in Methods.

**Power handling and coupling stability**. A chip-less test of the UV-bonded joint confirmed adhesive robustness up to 3 W (Fig. 4a), far above the ≲0.5 W limit of free-space coupling; accordingly, all subsequent experiments using the fibre-coupled microresonator were performed

with a 1 W pump. The direct fibre-to-chip connection achieved an initial coupling efficiency of 38.8%, and the coupling efficiency fluctuated by only ±0.09% over 10 h (Fig. 4b), indicating that the coupling efficiency remained essentially unchanged from its initial value. By contrast, the lensed-fibre set-up exhibited a ± 1.27 % fluctuation in the coupling efficiency, which varied the coupling efficiency from 50.0% to 43% during the same interval. The quasi-periodic ±1.27 % fluctuation (with a characteristic period of ~10 min) is attributed to weak mechanical vibrations transmitted through the optical table, likely originating from intermittently operating equipment (e.g., compressors) colocated on the same table, to which the free-space lensed-fibre coupling is particularly sensitive. This comparison highlights the superiority of the high-NA SMF interface in high-power tolerance and long-term stability, prerequisites for reliable, continuous soliton microcomb operation.

**Long-term soliton operation**. The direct fibre-coupled microresonator generated a 560 GHz soliton comb with a $sech^2$ envelope (Fig. 4c). Five passive trials yielded lifetimes of 95, 166, 63, 127, and 240 min (138 ± 69 min; mean ± s.d., $n = 5$); three of these runs (95, 63 and 127 min) were terminated manually owing to external lab constraints. In addition, a single long-term endurance test confirmed continuous soliton operation for 27.7 h, demonstrating the robustness of the fibre-coupled interface. An otherwise identical free-space, lensed-SMF interface produced a spectrally similar soliton comb (Fig. 4d) but sustained solitons for only 4.0 ± 0.13 min (mean ± s.d., $n = 5$). This shorter lifetime stems primarily from the lower pump power ( ≈ 0.5 W) and coupling drift caused by thermal motion of the lens tip, which together push the intracavity power below the soliton threshold. These results highlight the mechanical and thermal superiority of the high-NA SMF interface for sustained microcomb operation.

**THz wireless transmission in 560 GHz band**
**Transmitter**. Figure 5 sketches the 560 GHz wireless test-bed. A soliton microcomb generated in the fibre-coupled SiN resonator of Fig. 1 ($\lambda_0 = 1560.5$ nm, $f_{rep} = 560$ GHz, $P_{avg} \approx 1$ mW) is split, and two adjacent comb lines optical injection lock (OIL) a pair of high-power DFB lasers (DFB1 and DFB2), faithfully copying the comb's low phase noise while

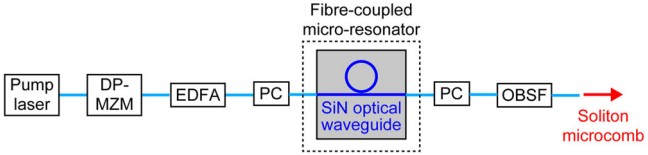

**Fig. 3 | Experimental setup for soliton microcomb generation in a fibre-coupled SiN microresonator.** A carrier-suppressed single-sideband (CS-SSB) pump is generated from an external-cavity pump laser using a dual-parallel Mach-Zehnder modulator (DP-MZM), amplified by an erbium-doped fibre amplifier (EDFA), and polarisation-optimised by a polarisation controller (PC) before coupling into the SiN microresonator via the high-NA fibre interface. Residual pump light is removed by an optical band-stop filter (OBSF), yielding the soliton microcomb output.

**Fig. 4 | Characterisation of coupling stability, power tolerance, and soliton performance.**
**a** Power-handling evaluation of the UV-bonded fibre interface without the microresonator chip. Output transmission showed negligible degradation at 1 W (red, −0.18%), 2 W (blue, −1.4%), and 3 W (green, −1.2%) input levels, confirming high thermal robustness. **b** Temporal stability of the coupling efficiency for the direct fibre-to-chip interface (red) and free-space lensed-fibre coupling (blue) over 10 h. The direct interface maintained a constant coupling efficiency of 38.8% (standard deviation ±0.09%), whereas the free-space counterpart suffered degradation from an initial efficiency of 50% down to as low as 43% due to thermal and mechanical instability (standard deviation ±1.27%). **c** Optical spectrum of a 560 GHz soliton microcomb generated using the direct fibre-coupled system. The measured spectrum is well described by a $sech^2$-shaped envelope, as confirmed by comparison with the fitted $sech^2$ curve (blue line). Stable soliton states persisted for 95–166 min (mean ± standard deviation = 113 ± 44 min). **d** Optical spectrum of a soliton microcomb generated with a lensed-fibre interface under identical conditions. Despite the comparable spectral shape, the soliton lifetime was much shorter, typically lasting only several minutes to around 20 minutes, due to pump instability.

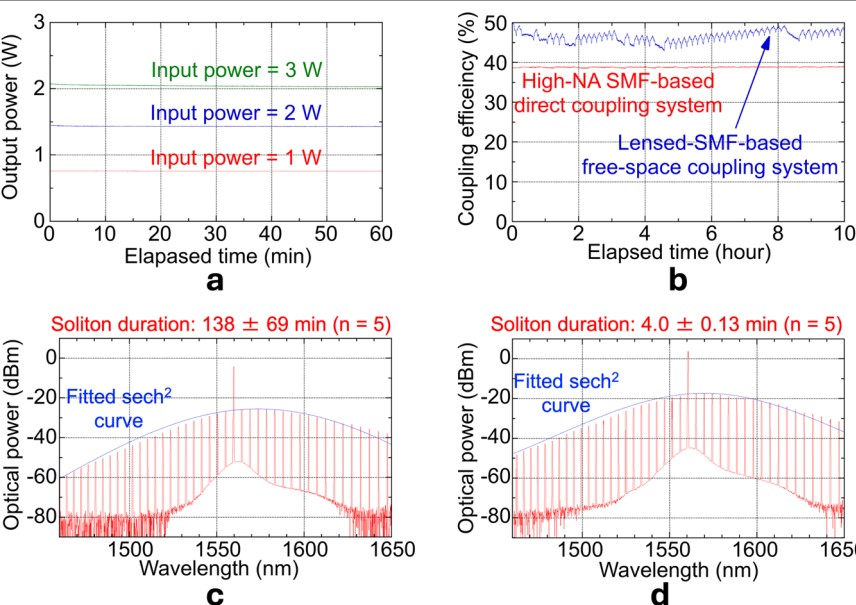

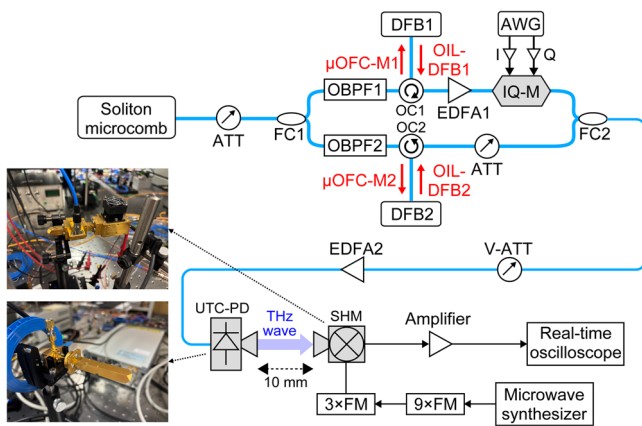

**Fig. 5 | Photonic-wireless transmission setup at 560 GHz based on soliton microcombs.** A detailed schematic of the 560 GHz wireless transmission system. A soliton microcomb ($\lambda_0$ = 1560.5 nm, $f_{rep}$ = 560 GHz, average power ≈1 mW) generated in the fibre-coupled SiN microresonator (Fig. 1) passes through an optical attenuator (ATT) and is split into two adjacent comb lines by a fibre coupler (FC1). The extracted microcomb line pairs (μOFC1-M1 and μOFC1-M2), selected using optical bandpass filters (OPBF1 and OPBF2), optically injection-lock (OIL) two high-power distributed-feedback lasers (DFB1 and DFB2) via optical circulators (OC1 and OC2). This process transfers the comb's low phase noise to the DFB lasers while enhancing optical signal-to-noise ratio (OSNR). One OIL-DFB output (OIL-DFB1) is modulated by an in-phase and quadrature modulator (IQ-M) driven by an arbitrary waveform generator (AWG), whereas the other (OIL-DFB2) remains unmodulated. The two optical signals are power-balanced using an erbium-doped fibre amplifier (EDFA1) and an additional ATT, recombined by a second fibre coupler (FC2), and photomixed in a uni-traveling-carrier photodiode (UTC-PD) to generate the 560 GHz carrier wave after passing through a variable optical attenuator (V-ATT) and a second erbium-doped fibre amplifier (EDFA2). After free-space propagation over a 10 mm path, the THz signal is coherently down-converted by a sub-harmonic mixer (SHM) and digitized for demodulation. The local-oscillator signal for the SHM is derived from a microwave synthesizer followed by a pair of frequency multipliers (3xFM and 9xFM). Insets show photographs of the UTC-PD and the SHM used in the experiment.

boosting optical signal-to-noise ratio (OSNR)[32]. One OIL-DFB (OIL-DFB1) is IQ-modulated with QPSK ($\leq$ 42 GBaud) or 16QAM symbols ($\leq$ 28 GBaud) whereas the other (OIL-DFB2) remains unmodulated. After power balancing, the two tones are photomixed in a waveguide-integrated UTC-PD[33], yielding a 560 GHz carrier of 3.2 μW that travels 10 mm in free space.

**Receiver.** At the receiver, a waveguide-integrated sub-harmonic mixer (SHM; IF = DC-40 GHz) driven by a ×27 frequency-multiplied microwave down-converts the THz signal; the intermediate-frequency output is amplified and digitised by a 40 GHz real-time oscilloscope for demodulation. All demodulation and error-vector magnitude (EVM) evaluation were performed using the oscilloscope's built-in real-time digital signal processing, without any additional offline equalization, frequency-offset compensation, or phase tracking. Component specifications are detailed in *Methods*.

**Performance metric.** Transmission quality was evaluated qualitatively by visually inspecting the symbol distribution in the received QPSK and 16QAM constellation diagrams and quantitatively by calculating the corresponding EVM from these diagrams. The link is regarded as error-free when the EVM remains below the HD-FEC threshold that corresponds to a bit error rate (BER) = $3.8 \times 10^{-3}$ (see *Methods* for details).

**QPSK transmission performance.** Constellations in Fig. 6a-e trace QPSK operation from 10 to 42 GBaud. The symbol rates for each modulation format were chosen by approximately dividing the practically

accessible transmission window into five evenly spaced operating points, in order to capture the evolution of transmission performance and dominant impairment mechanisms across the full usable range. At higher symbol rates, the IF chain, comprising the SHM, IF amplifiers, and oscilloscope frontend, effectively acts as a low-pass filter with a gradual roll-off, such that bandwidth limitations are inherently captured in the measured EVM. At the lowest rate the symbol clusters broaden chiefly along the phase axis, a hallmark of carrier-phase noise; once the rate exceeds 20 GBaud, the pattern balloons uniformly, signalling that additive noise has overtaken phase noise as the dominant impairment. This additive-noise-dominated regime originates from a gradual reduction in effective SNR at higher symbol rates, caused by the finite and non-flat frequency response of the IF chain, which integrates increased thermal and amplifier noise and introduces inter-symbol interference through amplitude and group-delay distortion. Quantitatively, every data point in Fig. 6f resides beneath the 37.5 % HD-FEC limit (BER = $3.8 \times 10^{-3}$). The two extrema illustrate the margin: 10 GBaud attains EVM = 10.6 % (BER $\lesssim 10^{-12}$, effectively error-free), whereas 42 GBaud records EVM = 35 % (BER $\approx 2.8 \times 10^{-3}$) yet still clears the HD-FEC hurdle. Taken together with the dual-bit payload of QPSK, these rates translate into a gross throughput of 84 Gbps, equivalent to a spectral efficiency of 2 bit $s^{-1}$ $Hz^{-1}$.

**16QAM transmission performance.** Figure 7a–e repeats the analysis for 16QAM at symbol rates spanning 5-28 GBaud. The 5-15 GBaud constellations are dominated by phase-driven angular smearing, whereas at 25 and 28 GBaud both inner and outer decision boundaries blur isotopically, indicating that SNR, not phase noise, now defines the performance ceiling, mirroring the behaviour observed for QPSK. Figure 7f shows that EVM remains confined within the 12.9 % HD-FEC limit up to the top-end rate: the link is HD-FEC-compatible at 28 GBaud (EVM = 12.9 %, BER $\approx 3.8 \times 10^{-3}$). With four bits per symbol, these figures yield a gross data rate of 112 Gbps, corresponding to 4 bit $s^{-1}$ $Hz^{-1}$ and confirming >100 Gbps wireless capacity from the microcomb-driven THz link.

## Discussion
### Benchmark beyond 350 GHz
Our demonstration of single-channel 112 Gbps-class wireless transmission at 560 GHz establishes a new benchmark for photonic THz systems operating beyond 350 GHz. Using a fibre-coupled soliton microcomb, OIL-based photomixing, and coherent IQ modulation and demodulation, we achieve ultra-high-capacity transmission with long-term stability. Despite achieving HD-FEC-qualified single-channel data rates above 150 Gbps in the 300 GHz band with 16QAM[22,24], previous microcomb-based systems nevertheless remained limited to carrier frequencies below 350 GHz, hindered by difficulties in high-frequency generation and phase stabilization. In particular, benchmarks for photonic THz transmission beyond 350 GHz are summarized in Table 1, which highlights that at 560 GHz prior demonstrations have been limited to 2 Gbps-class transmission, using simple modulation schemes (OOK or QPSK) and low symbol rates[25,26]. By combining high-speed coherent modulation and demodulation with sustained soliton microcomb operation, we overcome key barriers to high-capacity transmission at sub-THz frequencies, providing the phase stability and bandwidth needed for next-generation wireless backhaul.

### Phase-noise advantage of microcomb OIL
Building on this benchmark, we next address phase-noise suppression, a critical requirement for high-order modulation at sub-THz frequencies. The use of microcomb-based OIL improves the phase coherence of photomixed THz carriers. This improvement was first confirmed through RF spectral analysis of unmodulated 560 GHz carriers. As shown in Fig. 8a, the photomixing of the microcomb-OIL DFB system exhibited substantially narrower linewidths than that of the free-running DFB configuration, reflecting

**Fig. 6 | QPSK transmission performance of the microcomb-driven 560 GHz link. a–e** Received constellations at symbol rates of 10, 20, 30, 40 and 42 GBaud, respectively. Angular spreading of the points indicates carrier-phase noise, whereas isotropic dispersion signals an SNR-limited condition (see Evaluation methodology for transmission performance in *Methods*). **f** Error-vector magnitude (EVM) versus symbol rate. The green line marks the HD-FEC threshold of 37.5 % (BER = $3.8 \times 10^{-3}$); all measured points lie below this limit, confirming error-correctable QPSK operation up to 42 GBaud.

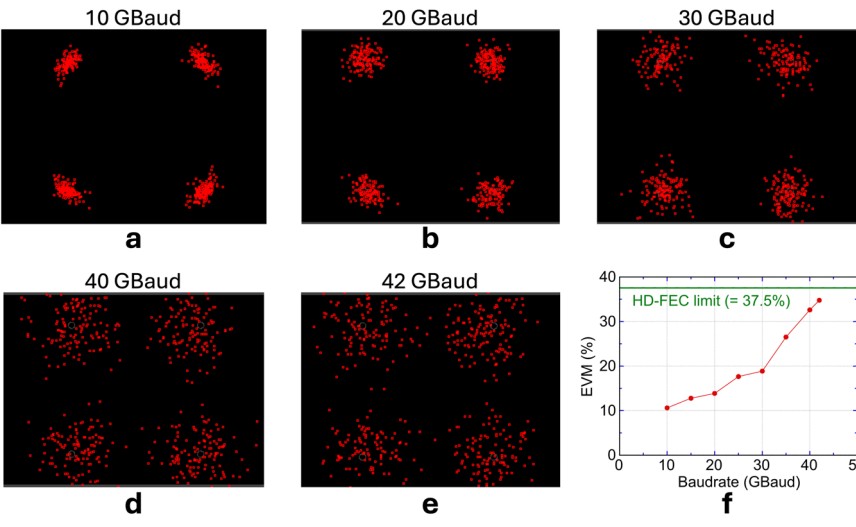

**Fig. 7 | 16QAM transmission performance of the microcomb-driven 560 GHz link. a-e** Received constellations at symbol rates of 5, 10, 15, 25 and 28 GBaud, respectively. Angular (phase) spreading characterises the lower-rate panels, whereas the more circular (isotropic) dispersion in the 25 GBaud and 28 GBaud maps is symptomatic of SNR-limited operation (see Methods for constellation diagnostics). **f** Error-vector magnitude (EVM, red circles) versus symbol rate. The green line marks the HD-FEC threshold of 12.9 % (BER = $3.8 \times 10^{-3}$). All measured points fall below this limit; at 28 GBaud the EVM is 12.9 %, within margin and corresponding to a gross throughput of 112 Gbps.

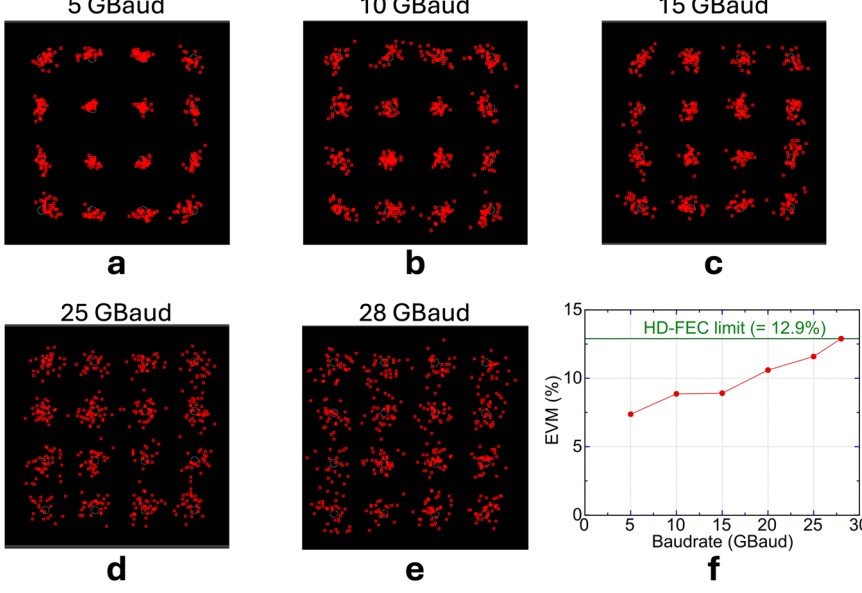

the successful transfer of low phase noise from the microcomb. The detailed phase-noise characteristics of microcomb-based OIL have been quantitatively analysed in our previous work[23]. This enhanced carrier coherence directly contributes to improved modulation stability and reduced phase noise impact. A side-by-side test with identical THz hardware compared microcomb-OIL DFBs with free-running DFBs. At 10 GBaud 16QAM (Fig. 8b, c) the microcomb-OIL link shows a tighter constellation and EVM = 8.9 % (BER ≈ $6 \times 10^{-5}$) versus 9.8 % ($2 \times 10^{-4}$) for the free-running pair, demonstrating the benefit of phase-noise suppression. At 25 GBaud (Fig. 8d, e), SNR, rather than phase noise, dominates; even so, the OIL link remains inside the HD-FEC limit with EVM = 11.6 %, whereas the free-running link rises to 13.2% and fails the margin. These results confirm that microcomb-based OIL enhances phase-noise-limited transmission quality, particularly at lower to intermediate baud rates, while maintaining performance advantages even under higher symbol rate conditions. Because the received signals are processed using real-time carrier recovery and phase tracking in the oscilloscope, a substantial fraction of low-frequency phase fluctuations is compensated, so that the phase-noise suppression evident in the RF spectrum does not translate proportionally into constellation tightening.

**Operational robustness of fibre packaging**

Our fibre-coupled packaging, using high-NA SMF and UV-bonded interfaces, provides stable soliton microcomb operation exceeding 24 h. This contrasts sharply with conventional free-space coupling, typically limited to several minutes due to thermal drift and coupling degradation. Such thermal and mechanical robustness makes the approach directly deployable in real-world THz transceivers.

**Path toward extended transmission reach**

While our system prioritizes stable generation of soliton microcombs with a 560 GHz repetition rate, the corresponding THz carrier frequency lies near an atmospheric water vapour absorption line, leading to severe attenuation (7.1 dB/m)[34,35]. The present demonstration therefore serves as a proof-of-concept at 560 GHz, while future deployment will realistically target lower-attenuation bands such as 500 GHz (0.041 dB/m)[34,35], enabling metre-scale transmission. To assess possible strategies for extending reach, we simulated EVM degradation in 25 GBaud 16QAM transmission as a function of propagation distance for carrier frequencies of 560 GHz and 500 GHz with a launched THz power of 3 μW, using antennas with gains of 25 dBi at 560 GHz and gains of 26 dBi at 500 GHz

**Table 1 | Reported photonic wireless transmissions beyond 350 GHz including the present work as the first 100 Gbps-class demonstration at 560 GHz**

| Frequency band (GHz) | Channel number | Modulation Baud-rate | Gross bit rate per channel (Gbps) | Distance (m) | Frequency reference | Transmitter Receiver | Reference |
|---|---|---|---|---|---|---|---|
| 350 | 1 | 16QAM 25 GBaud | 100 | 2 | Free-running lasers | UTC-PD Heterodyne detection | 37 |
| 408 | 1 (OFDM) | 16QAM 32 GBaud | 157 | 10.7 | MLL-OFC | UTC-PD Heterodyne detection | 14 |
| 425 | 1 | 16QAM 32 GBaud | 128 | 0.5 | EOM-OFC | UTC-PD Heterodyne detection | 38 |
| 325-475 | 6 (WDM) | 16QAM 12.5 GBaud | 50 | 0.5 | EOM-OFC | UTC-PD Heterodyne detection | 39 |
| 300-500 | 8 (WDM) | QPSK 10 GBaud | 20 | 0.5 | EOM-OFC | UTC-PD Heterodyne detection | 40 |
| 560 | 1 | OOK 2 GBaud | 2 | 0.6 | Microcomb | UTC-PD Direct detection | 25 |
| 560 | 1 | QPSK 1 GBaud | 2 | 0.01 | Microcomb | UTC-PD Direct detection | 26 |
| 500-724 | 14 (WDM) | 32QAM 40 GBaud | 160 | Back-to-back | Free-running lasers | UTC-PD Schottky mixer | 8 |
| 560 | 1 | 16QAM 28 GBaud | 112 | 0.01 | Microcomb | UTC-PD Heterodyne detection | Present work |

**Fig. 8 | Phase-noise advantage of microcomb-based optical injection locking (OIL). a** RF spectra of unmodulated 560 GHz carriers generated by photomixing in a DFB laser pair for the OIL-driven (red) and free-running (blue) configurations. **b,c** Received 10 GBaud 16QAM constellations for the OIL-driven and free-running cases. **d, e** Received 25 GBaud 16QAM constellations for the same configurations. The OIL scheme produces a narrower RF linewidth and tighter constellations, with performance remaining within the HD-FEC threshold at both symbol rates.

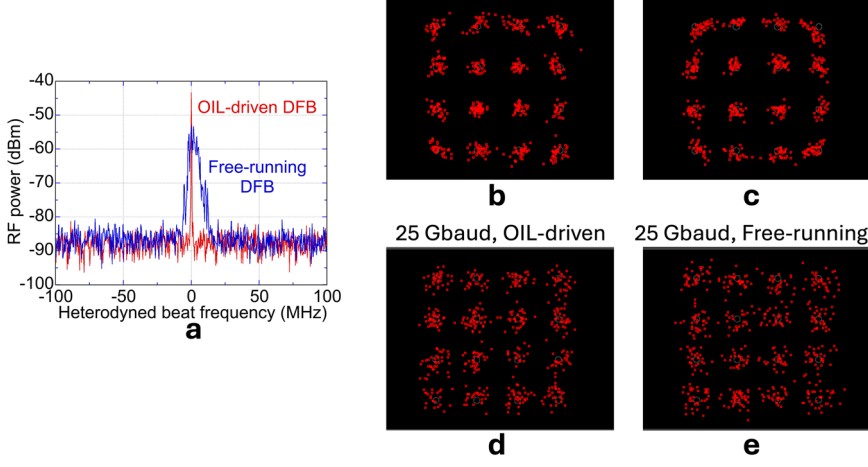

(see black and orange traces in Fig. 9). The experimentally measured data point (EVM = 11.6% at a propagation distance of 10 mm) is overlaid in Fig. 9 as a purple star for direct comparison, showing good qualitative agreement with the simulated distance-dependent trend and confirming that the simulation provides a conservative projection of transmission reach rather than an exact reproduction of the experimental condition. While the system sustains HD-FEC-compliant transmission up to around 35 mm at 560 GHz, consistent with experiments, shifting the carrier to 500 GHz extends the transmission range to approximately 55 mm. In parallel, increasing THz output power using next-generation high-power UTC-PDs[10], such as SiC-based UTC-PDs[36], modified UTC-PDs [10], and array-integrated UTC-PDs[10,36], offers a complementary approach to extending transmission distances. We further evaluated the propagation-distance dependence of EVM for a 500 GHz carrier when the launched THz power was increased from 3 µW to 30 µW, as shown by the blue trace in Fig. 9, achieving HD-FEC-qualified transmission up to 173 mm. From the perspective of further extending THz propagation distance, we evaluated the use of high-gain lens horn antennas for both THz generation and detection. For a 500 GHz carrier at 30 µW, increasing the antenna gain from 26 dBi to 51 dBi, within the range of

commercially available products, markedly reduced free-space path loss, as shown by the red trace in Fig. 9, enabling error-free operation over distances up to 45 m.

Together, these advances chart a clear and commercially realistic path from today's 112 Gbps, 10 mm proof-of-concept toward metre- to tens-of-metre-scale, multi-100-Gbps THz backhaul systems using advanced QAM formats in the unallocated THz spectrum, providing a tangible physical-layer foundation for next-generation 6 G wireless infrastructure.

## Methods
### Fibre-coupled SiN microresonator packaging
We employed a custom-designed SiN ring microresonator (LIGENTEC S.A.) featuring FSR of 560 GHz and a Q factor of approximately $10^6$. The top part of Fig. 1 illustrates a schematic view of the implemented direct fibre-to-chip connection configuration. For the optical interface, a standard SMF (SMF; cladding diameter = 125 µm, coating diameter = 250 µm, MFD = 10.5 µm at 1.55 µm) terminated with an FC/APC connector was fusion-spliced to a high-NA SMF (cladding = 125 µm, coating = 250 µm, MFD = 3.2 µm at 1.55 µm). This was done using

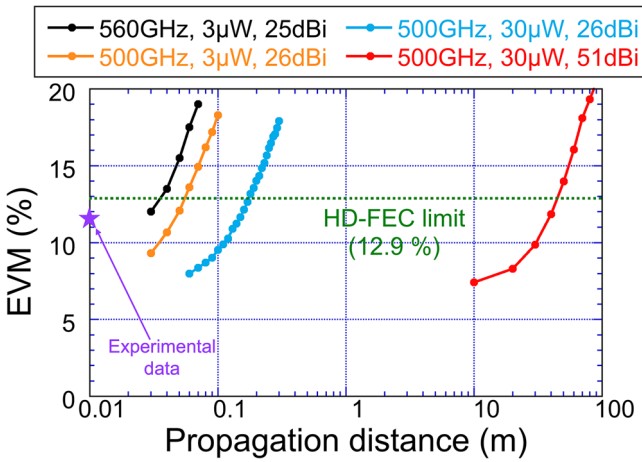

**Fig. 9 | Simulated EVM versus propagation distance for different carrier frequencies, THz output powers, and antenna gains.** Performance projections for 25 GBaud 16QAM transmission were obtained using OptiSystem. The experimentally measured data point (purple star) corresponds to EVM = 11.6% at a propagation distance of 10 mm under a launched THz power of 3.2 µW. *Black trace*: carrier frequency = 560 GHz, THz power = 3 µW, antenna gain = 25 dBi, far-field boundary = 172 mm, atmospheric attenuation = 7.1 dB/m. *Orange trace*: carrier frequency = 500 GHz, THz power = 3 µW, antenna gain = 26 dBi, far-field boundary = 256 mm, atmospheric attenuation = 0.041 dB/m. *Blue trace*: carrier frequency = 500 GHz, THz power = 30 µW, antenna gain = 26 dBi, far-field boundary = 256 mm, atmospheric attenuation = 0.041 dB/m. *Red trace*: carrier frequency = 500 GHz, THz power = 30 µW, antenna gain = 51 dBi, far-field boundary = 161 m, atmospheric attenuation = 0.041 dB/m.

thermoelectric-controlled (TEC) fusion splicing (loss <0.3 dB) to optimize MFD matching. The high-NA SMF was directly coupled to the input waveguide (0.8 µm height × 1.0 µm width) of the SiN microresonator via a single-core fibre array (FA) mounted on a Tempax glass plate for mechanical reinforcement, as illustrated in the bottom part of Fig. 1. To secure the fibre-chip interface, we used a UV-curable optical adhesive, which primarily provided permanent mechanical fixation while also helping to minimize refractive-index mismatch and suppress optical loss. Both the fibre tip and the waveguide facet were optically polished, and precise alignment was performed before applying and curing the adhesive under UV exposure. This method yielded robust and efficient fibre-to-chip coupling, ensuring both mechanical stability and optical reliability. An identical optical configuration was applied to the output port, comprising a single-core FA, a high-NA SMF, and a standard SMF terminated with an FC/APC connector. Under these conditions, an input power of 480 mW and an output power of 186 mW correspond to a coupling efficiency of 38.8%, yielding a total insertion loss of approximately 4.1 dB. Accounting for the FC/APC connector pair loss ($\approx$ 0.2–0.3 dB), the SMF-to-high-NA-SMF fusion splice losses ($<$ 0.3 dB each), and negligible propagation losses in the fibres and on-chip waveguides, the remaining loss is attributed to the fibre-to-chip interfaces, giving an estimated facet loss of approximately 1.6 dB per facet (i.e., for both input and output interfaces). For comparison, the lensed-fibre free-space coupling system exhibited an input power of 480 mW and an output power of 240 mW, corresponding to a coupling efficiency of 50% and a total insertion loss of approximately 3.0 dB. After accounting for the FC/APC connector loss ($\approx$ 0.2-0.3 dB) and negligible propagation losses in the fibres and on-chip waveguides, the remaining loss is attributed to the fibre-to-chip interfaces, giving an estimated facet loss of approximately 1.4 dB per facet. Thus, the two approaches represent a clear trade-off between initial coupling efficiency and long-term stability, with the high-NA fibre-coupled interface favoring robust, high-power, continuous soliton microcomb operation.

Figure 2 presents two photographs comparing (a) the direct fibre-to-chip coupling configuration developed in this study with (b) a conventional

free-space coupling setup using a lensed SMF. The side-by-side comparison highlights the substantial reduction in system footprint achieved by our approach, demonstrating its potential for compact and scalable THz transceiver integration.

## Generation of soliton microcombs
Soliton microcombs were generated by launching pump light through the high-NA SMF-based direct fibre-to-chip connection system, as shown in Fig. 3. The pump source was an external-cavity laser diode with a linewidth of 500 kHz and output power of 20 mW, operating at a centre wavelength of 1559.75 nm. The laser output was modulated using a dual-parallel Mach-Zehnder modulator (DP-MZM) to generate a CS-SSB signal. The modulated optical output ($\sim$ 1.4 mW) was then amplified by a home-built erbium-doped fibre amplifier (EDFA). After amplification, the signal was injected into the input SMF of the SiN microresonator. A polarization controller (PC) was used to optimize the polarization state for maximum coupling efficiency. To initiate the soliton state, the CW laser wavelength was rapidly swept from the short-wavelength side toward the resonance of the microresonator[27,28]. Residual pump light was subsequently removed using OBSF. For reference, we also evaluated a conventional free-space coupling configuration employing a lensed SMF, as detailed in our previous study[26]. Apart from the coupling of the pump light into the microresonator, the rest of the experimental setup remained unchanged from the main configuration.

## High-power endurance test of the UV-bonded joint
To isolate the optical adhesive, the microresonator chip was removed and the two high-NA SMFs were butt-coupled and bonded with the same UV-curable adhesive. An EDFA delivered 1, 2, or 3 W of 1550 nm light; launch power was monitored with a 1 % tap coupler. Transmitted power was measured at 1 Hz for 60 min. These traces are plotted in Fig. 4a. The fractional loss change, serving as a direct indicator of the fibre-to-waveguide coupling loss, was 0.18 % (1 W), 1.4 % (2 W) and 1.2 % (3 W), indicating no measurable adhesive degradation. Free-space coupling with a lensed SMF, by contrast, became thermally unstable above 0.5 W because local heating of the lens tip induced beam-pointing wobble and periodic de-alignment. We therefore capped the pump power at 0.5 W for the reference setup, while operating the direct fibre interface at 1 W to exploit its higher power tolerance.

## Coupling-stability measurement
A SiN microresonator equipped with the direct high-NA SMF interface (Fig. 1) was driven by a continuous-wave pump whose wavelength was detuned from the microresonator's resonance mode so that no soliton was generated. Optical power $P_{in}$ at the fibre input (high-NA SMF side) and $P_{out}$ at the output connector were recorded with calibrated power meters. The initial values were $P_{in}$ = 480 mW and $P_{out}$ = 186 mW, giving a coupling efficiency $\eta_0 = P_{out} / P_{in}$ = 38.8 %. The coupling efficiency was logged every 1 s for 10 h in an experimental room held at 19.8 ± 0.3 °C. The resulting temporal traces are plotted in Fig. 4b. The same protocol was repeated with a lensed-SMF free-space setup; here the initial efficiency was 50.0 % ($P_{in}$ = 480 mW, $P_{out}$ = 240 mW). The lower initial coupling efficiency of the direct high-NA SMF interface compared with the lensed-SMF configuration arises from fixed (non-adjustable) mode-field matching between the high-NA SMF and the inverse-tapered access waveguide, as well as diffraction and beam expansion in the bonded interface, whereas the lensed-SMF allows finer beam-size adjustment through active alignment. Standard deviations of efficiency fluctuation (± 0.09 % for the direct interface, ±1.27 % for the lensed-SMF) were used to quantify coupling stability.

## THz wireless transmission system
We employed a soliton microcomb source based on the high-NA SMF direct coupling scheme described in Fig. 1. The generated soliton microcomb had a centre wavelength of 1560.5 nm, a repetition rate of

560 GHz, and an average output power of 1 mW. In order to transfer the favourable phase noise characteristics of the soliton microcomb to the two-wavelength laser light used for photomixing, we applied OIL of two high-power DFBs to two adjacent microcomb lines[32]. This approach enabled selective amplification of the comb lines while preserving their low phase noise, and simultaneously improved the OSNR by suppressing the amplified spontaneous emission (ASE) background.

A detailed schematic of the experimental setup is shown in Fig. 5. The microcomb output was split into two branches via a 50:50 fibre coupler (FC1) after passing through an optical attenuator (ATT). Each branch was filtered using tunable ultra-narrowband optical bandpass filters (OBPF1 and OBPF2; Alnair Labs, TFF-15-1-PM-L-100-FA, passband = 1520–1580 nm). Two adjacent comb lines were extracted: one at 1551.6 nm (μOFC-M1; 193.21 THz) and the other at 1547.2 nm (μOFC-M2; 193.77 THz), with output powers of approximately 1 μW each. These comb lines were used as master signals for OIL of two DFBs. The DFB slave lasers (DFB1: Gooch & Housego, AA1408-193200-100-PM250-FCA-NA, λ = 1551.6 nm, ν = 193.21 THz; DFB2: Gooch & Housego, AA1408-193700-100-PM900-FCA-NA, λ = 1547.2 nm, ν = 193.77 THz) had a line-width of 1 MHz and an output power of several tens mW to 100 mW. Each was optically injected with the corresponding comb line via an optical circulator (OC1 and OC2). OIL was achieved by fine-tuning the laser drive current so that the slave DFBs' frequencies fell within their locking range (a few hundred MHz). Successful locking was confirmed by detecting the beat signal between the injected comb line and DFB output using a photodetector and verifying its frequency convergence toward DC in an RF spectrum analyser.

The output from μOFC-M1-locked DFB1 (OIL-DFB1) was amplified to ~18 dBm using an erbium-doped fibre amplifier (EDFA1) to compensate for the optical insertion loss associated with the following IQ modulator (IQ-M). The amplified signal was then modulated using a high-speed IQ-M (ID Photonics, OMFT-C-00-FA; insertion loss <15.5 dB, extinction ratio = 25 dB, optical bandwidth = 45 GHz). Modulation signals were generated by an arbitrary waveform generator (AWG; Keysight M8196A, 92 GSa/s, 32 GHz bandwidth) and applied differentially. Two modulation formats were used: QPSK (up to 42 GBaud) and 16QAM (up to 28 GBaud). The μOFC-M2-locked DFB2 (OIL-DFB2), used as the unmodulated optical carrier, was attenuated to ~3 mW and bypassed the IQ-M. The modulated OIL-DFB1 and unmodulated OIL-DFB2 lights were combined using a fibre coupler (FC2), power-balanced via an optical variable attenuator (V-ATT: Japan Device, MVOA-1550-P5-2-B-Q-P3), and subsequently amplified with an additional EDFA (EDFA2) to achieve a total optical input power of 30 mW. The combined light was then launched into a waveguide-integrated UTC-PD (bandwidth = 470-810 GHz)[33]. Through photomixing, a 560 GHz THz wave was generated, with an output power of 3.2 μW. A standard diagonal horn antenna (Virginia Diodes, WM-380 (WR-1.5), freq. = 500–750 GHz, gain = 25 dBi at 560 GHz), integrated with the UTC-PD, was used to radiate the THz signal into free space. The THz wave propagated through free space over a 10 mm path.

At the receiver, the incoming THz wave was collected by an identical diagonal horn antenna (Virginia Diodes, WM-380 (WR-1.5), freq. = 500–750 GHz, gain = 25 dBi at 560 GHz) and coupled into a waveguide-integrated SHM (Virginia Diodes, WR1.5SHM, RF = 500-750 GHz, LO = 250–375 GHz, IF = DC-40 GHz). An output signal from a microwave frequency synthesizer (Agilent E8257D, output frequency = 250 kHz–20 GHz) was multiplied by a 9× frequency multiplier (9×FM: VDI WR9.0 AMC-I, frequency bandwidth = 82–125 GHz) and subsequently by a 3× multiplier (3×FM: VDI WR2.8X3UHP, frequency bandwidth = 250–375 GHz), resulting in a total multiplication factor of 27 for SHM local oscillator (LO) feeding. The intermediate-frequency (IF) output was amplified by a power amplifier (SHF Communications, SHF S824 A, 80 kHz-35 GHz, gain = 25 dB), and finally digitized using a real-time oscilloscope (Keysight UXR0402AP, 40 GHz bandwidth, 256 GSa/s sampling rate).

## Evaluation methodology for transmission performance

To assess the transmission performance of the THz wireless link, we employed constellation diagrams, which map received modulated signal points onto the complex plane and offer an intuitive visualization of modulation quality. This representation is particularly useful for identifying key degradation factors: (a) phase noise, which causes angular spreading of signal points around their ideal positions, and (b) low SNR, which leads to omnidirectional dispersion due to additive white Gaussian noise, blurring symbol boundaries and reducing demodulation accuracy. Such visual diagnostics are especially informative for evaluating higher-order modulation schemes.

To complement this visual assessment, we used EVM as a quantitative metric. Although BER directly measures demodulation performance, it requires long measurement times and large datasets. In contrast, EVM offers a faster and more practical evaluation by quantifying the deviation between measured and ideal signal points. EVM and BER are closely related, and EVM thresholds can be used as effective surrogates for BER in many cases. In this study, EVM was calculated from each constellation diagram and compared against the HD-FEC threshold, which corresponds to a BER of $3.8 \times 10^{-3}$. This threshold is widely used as a benchmark in simplified system designs.

## Simulation framework for propagation-distance-dependent EVM considering carrier frequency, THz power, and antenna gain

All performance projections were obtained with OptiSystem 22.1 (Optiwave Systems Inc.), using the Friis transmission equation, the standard formulation for predicting received power under free-space propagation. A 25 GBaud 16QAM baseband stream was mapped onto one of two optical carriers that emulate adjacent lines of the soliton microcomb. To replicate the comb's mutual coherence, the two continuous-wave lasers were assigned identical phase excursions (single-sideband linewidth = 0.5 MHz), ensuring common-mode phase noise. After IQ modulation, the optical tones were photomixed in a waveguide-integrated UTC-PD. A horn antenna (gain = 25 dBi at 560 GHz and 26 dBi at 500 GHz) or a high-gain lens horn antennas (gain = 51 dBi at 500 GHz), attached to the UTC-PD, was used to radiate the THz carriers (frequency = 560 GHz or 500 GHz; power = 3 μW or 30 μW), which then propagated through free space. Free-space path loss was included in the analysis, with the far-field condition taken as 172 mm at 560 GHz with a 25 dBi antenna, 256 mm at 500 GHz with a 26 dBi antenna, and 161 m at 500 GHz with a 51 dBi antenna. Furthermore, atmospheric attenuation was included, with losses of 7.1 dB/m at 560 GHz and 0.041 dB/m at 500 GHz[34,35]. At the receiver, a waveguide-integrated SHM front-ended by a horn antenna (gain = 25 dBi at 560 GHz and 26 dBi at 500 GHz) or a high-gain lens horn antennas (gain = 51 dBi at 500 GHz) down-converted the signal for digital processing. EVM was calculated using OptiSystem's standard 16QAM demodulation library, and distance-dependent EVM penalties were evaluated by comparing carrier frequencies of 560 GHz and 500 GHz, THz output powers of 3 μW and 30 μW, and antenna gains of 25 dBi, 26 dBi, and 51 dBi, as summarized in Fig. 9.

## Data availability

Data underlying the results presented in this paper are not publicly available at this time but may be obtained from the authors upon reasonable request.

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

## Acknowledgements

This work was supported in part by the Ministry of Internal Affairs and Communications (MIC) of Japan through the FORWARD program (JPMI240910001) and the R&D project High-speed THz communication based on radio and optical direct conversion" (JPJ000254) also funded by MIC. Additional support was provided by the Cabinet Office of the Government of Japan (Promotion of Regional Industries and Universities), Tokushima Prefecture (Creation and Application of Next-Generation Photonics), and the Regional Innovation and Excellence (J-PEAKS): Program for Promoting the Enhancement of Research Universities as Regional Centres of Knowledge by Japan Society for the Promotion of Science (JSPS), Grant Number JPJS00420240022. This work was also supported by JSPS KAKENHI (24K21237). We would like to express our sincere gratitude to Drs. Shuichiro Asakawa and Naomi Kawakami of NTT Advanced Technology Corporation for their support regarding the SMF-based direct connection systems of microresonator.

## Author contributions

T.Y. and Y.T. conceived and designed the project. Y.T., T.K., M.T., Y.M., K.Y., K.N., N.K. and T.N. performed the experiments. Y.T., H.K., T.K., M.T., Y.O., E.H., I.M., A.K., S.H., N.K. and T.N. analyzed the data and performed numerical simulations. T.Y., Y.T. and H.K. wrote the manuscript, and all authors reviewed and edited the article. T.Y. supervised the project and acquired funding.

## Competing interests

The authors declare no competing interests.
