## [Transparent Peer Review file · Communications Engineering]

Beyond 350 GHz: Single-channel 112 Gbps photonic wireless transmission at 560 GHz using soliton microcombs

Corresponding Author: Professor Takeshi Yasui

Version 0:

Reviewer comments:

Reviewer #1

(Remarks to the Author)

This manuscript presents a compact and robust SiN micro-ring resonator assembly for generating an optical frequency comb with a frequency spacing of 560 GHz. The mechanical bonding of input and output fibers using UV adhesive improves stability and extends operational lifetime. Two DFB lasers are locked to two adjacent comb lines to boost the output power, then used to generate a modulated wireless signal by photomixing in a UTC photodiode. The generated 560 GHz signal is used for THz transmission, achieving a data rate of 112 Gbit/s.

The main novelty of the paper is the improved packaging of the microcomb source which enables stable soliton operation for longer times (more than 24 hours), which clearly makes communication systems experiments more practical, but the stability would still be insufficient for use in a commercial communication system. The authors have also significantly increased the data transmission rate compared to their previous work using the microcomb.

This is impressive and interesting work, but we feel several aspects of the paper need clarification before it could be considered suitable for publication.

Our comments are as follows:

1. Some further details on the soliton microcomb should be given. (a) Other than the micro-ring resonator being fabricated on a SiN platform, no details of the design or dimensions are given. This should be addressed. (b) The paper refers to CS-SSB pump light being “rapidly swept” to initiate soliton formation. The method of generating the CS-SSB signal should be stated and “rapidly” should be quantified. (c) It is stated that the comb spectrum has a sech^2 -shaped envelope, but a fitting curve is not presented, so it is difficult to judge how closely the spectrum matches a sech^2 shape. Fitting to the spectrum would also allow the soliton pulse duration to be estimated. (d) The pump power used is 1 W, but the total comb power is only 1 mW, and individual filtered lines are only 1 μ W. The authors might wish to comment on the power efficiency of the approach.
2. Please clarify the IF frequency. The operating frequency range of the SHM is DC–40 GHz and the bandwidth of the real-time scope is also 40 GHz, whereas the demonstrated 42 Gbaud QPSK link already exceeds this bandwidth limit. It would be helpful to understand how the IF signal performs beyond 40 GHz. Additional details on the received signals would therefore be helpful, possibly including the spectrum of the received IF signal.
3. The offline post processing of the IF signal captured on the real-time oscilloscope should be described, including whether any equalization and frequency and phase tracking is employed.
4. It would be helpful if the BER performance could be measured, as this is a very basic and widely accepted performance metric. In addition, a comparison between the EVM and BER results would further strengthen the reliability of the presented performance evaluation. The duration of the samples taken using the real-time oscilloscope should also be stated, as this determines the minimum BER that can be inferred.
5. Please clarify what contributes to the additive noise for the QPSK transmission performance. More detailed noise analysis

would be beneficial. Furthermore, the choice of symbol rates needs further explanation: QPSK is evaluated at 10, 20, 30, 40, and 42 Gbaud, while 16-QAM is tested at 5, 10, 15, 25, and 28 Gbaud. Using identical symbol rates across different modulation formats would enable direct performance comparison and cross-validation. For example, the manuscript states that phase noise dominates at 10 Gbaud and additive noise begins to dominate from 20 Gbaud in the QPSK link. Observing similar trends in the 16-QAM case would significantly strengthen the analysis.

6. A more detailed comparison with the studies reported in references [25] and [26] would be valuable. The main difference appears to be the compact and robust SiN microcomb, which improves system stability. The 560 GHz carrier power in this work is 3.2 μ W, which is similar with the 3 μ W reported in [26], and the transmission path length is the same (10 mm). However, the data rate in [26] is limited to 1 Gbaud for QPSK with an EVM of 23.6%. It would therefore be helpful if the authors could clarify why a significantly higher data rate is achievable in the present work, and what fundamental system-level improvements contribute to this performance enhancement.

7. In the Discussion, it is stated that "frequency-agile transmission" has been achieved. The authors should explain and justify this statement. Isn't the frequency fixed by the FSR of the micro-ring resonator? The paper does not demonstrate that this can be adjusted.

8. Extended Data Figure 3(a) shows a significant improvement in phase noise with the DFB lasers locked to the microcomb lines. However, the EVM improvement shown in constellation diagrams in the rest of that figure is relatively modest. Can the authors comment further on this? Can the residual phase noise of the optical injection locked system be quantified?

9. Other minor points: (a) In the Introduction, it is stated that free-space path loss scales with frequency. Significantly, and to be precise, it scales with the square of frequency, as the authors are no doubt aware. (b) It is stated that the spectral efficiency with QPSK is 4 bit/s/Hz and with 16QAM it is 8 bit/s/Hz. Considering the optical or wireless passband bandwidth, these values should be 2 bit/s/Hz and 4 bits/s/Hz respectively. (c) Please add reference numbers to Extended Data Table 1 so that readers can more easily locate the corresponding sources.

Reviewer #2

(Remarks to the Author)

The manuscript presents an experimental demonstration of high-speed wireless transmission at 560 GHz, employing a fiber-packaged silicon-nitride soliton microcomb as a compact, low-phase-noise optical frequency reference. The authors achieve a single-channel data throughput of up to 112 Gbps using QPSK and 16QAM modulation, establishing a record transmission rate in the 560 GHz band. The work addresses important challenges associated with next-generation THz wireless backhaul for 6G networks and demonstrates a tangible proof-of-concept for realizing >100 Gbps-class wireless links above 350 GHz.

The paper is in good structure, and the English is well written.

However, besides the novelty and results, the manuscript requires major revisions for clarity, to ensure the impact and reproducibility of the work. To guide the authors in this process, I list several more specific comments to be addressed (see attached document with specific comments).

Reviewer #3

(Remarks to the Author)

I co-reviewed this manuscript with one of the reviewers who provided the listed reports. This is part of the Communications Engineering initiative to facilitate training in peer review and to provide appropriate recognition for Early Career Researchers who co-review manuscripts.

Version 1:

Reviewer comments:

Reviewer #1

(Remarks to the Author)

The authors have provided a very detailed response to the comments in our review and that of the other reviewer. They have made significant and appropriate changes to the manuscript. We believe the paper is now suitable for publication.

Reviewer #2

(Remarks to the Author)

The authors have provided a comprehensive and well structured revision that addresses all comments raised during the initial review round.

The additional clarifications on the coupling evaluation of the soliton microcomb, signal processing chain, and comparative analysis with prior THz transmission demonstrations have significantly improved the technical transparency and reproducibility of the work. The updated figures and expanded explanations strengthen the interpretation of the results and enhance the overall coherence of the manuscript.

I have no further technical concerns. The manuscript is suitable for publication in its current form.

Reviewer #3

(Remarks to the Author)

I co-reviewed this manuscript with one of the reviewers who provided the listed reports. This is part of the Communications Engineering initiative to facilitate training in peer review and to provide appropriate recognition for Early Career Researchers who co-review manuscripts.

Response to Reviewer 1's comment

This manuscript presents a compact and robust SiN micro-ring resonator assembly for generating an optical frequency comb with a frequency spacing of 560 GHz. The mechanical bonding of input and output fibers using UV adhesive improves stability and extends operational lifetime. Two DFB lasers are locked to two adjacent comb lines to boost the output power, then used to generate a modulated wireless signal by photomixing in a UTC photodiode. The generated 560 GHz signal is used for THz transmission, achieving a data rate of 112 Gbit/s. The main novelty of the paper is the improved packaging of the microcomb source which enables stable soliton operation for longer times (more than 24 hours), which clearly makes communication systems experiments more practical, but the stability would still be insufficient for use in a commercial communication system. The authors have also significantly increased the data transmission rate compared to their previous work using the microcomb.

This is impressive and interesting work, but we feel several aspects of the paper need clarification before it could be considered suitable for publication.

We sincerely thank the reviewer for the careful reading of our manuscript and for the positive and constructive evaluation of our work. We greatly appreciate the recognition of the improved microcomb packaging, the enhanced operational stability, and the significant increase in the achieved data transmission rate compared to our previous work. We also acknowledge the reviewer's important concerns regarding the remaining limitations and the need for further clarification. In response, we have carefully addressed all comments point by point and revised the manuscript accordingly to improve clarity and completeness. All modifications are highlighted in red in the revised manuscript for ease of review.

1. Some further details on the soliton microcomb should be given. (a) Other than the micro-ring resonator being fabricated on a SiN platform, no details of the design or dimensions are given. This should be addressed.

We thank the reviewer for pointing out the need for additional details regarding the soliton microcomb source. The SiN micro-ring resonator used in this work was a custom-fabricated device, for which only the target free spectral range (FSR) was specified at the time of design. Other detailed design parameters and geometrical dimensions (e.g., waveguide cross section, ring radius, and coupling gap) are proprietary information of the manufacturer and are therefore not publicly disclosed. To clarify this point and avoid ambiguity, we have revised the manuscript to explicitly state that the micro-ring resonator was a custom device supplied by the manufacturer, and we have added the manufacturer name in the revised version. These clarifications have been included to improve transparency and reproducibility to the extent possible. See **Line 126**.

(b) The paper refers to CS-SSB pump light being “rapidly swept” to initiate soliton formation. The method of generating the CS-SSB signal should be stated and “rapidly” should be quantified.

We thank the reviewer for pointing out that the description of the CS-SSB pump sweeping procedure required further clarification. In response to this comment, we have revised the manuscript to explicitly describe both the generation method of the CS-SSB pump light and the quantitative sweep speed. Specifically, the CS-SSB signal was generated using a dual-parallel Mach–Zehnder modulator (DP-MZM). The modulation frequency of the DP-MZM was linearly swept using a voltage-controlled oscillator, resulting in an effective optical frequency sweep rate of approximately 100 PHz/s, which enabled rapid traversal of the cavity resonance and reliable initiation of soliton formation. Accordingly, the previously qualitative expression “rapidly swept” has now been replaced by a quantitative description of the sweep speed. See Lines 139-143 and 145-147.

(c) It is stated that the comb spectrum has a sech^2 -shaped envelope, but a fitting curve is not presented, so it is difficult to judge how closely the spectrum matches a sech^2 shape. Fitting to the spectrum would also allow the soliton pulse duration to be estimated.

We thank the reviewer for this helpful suggestion. In response to this comment, we have performed a sech^2 fitting to the measured optical spectrum of the soliton microcomb. The fitting results have now been added to Figs. 2c and 2d in the revised manuscript. These additions allow a clearer and more quantitative assessment of how closely the measured spectrum follows the expected sech^2 profile. See Figs. 2c and 2d and Line 654-656.

(d) The pump power used is 1 W, but the total comb power is only 1 mW, and individual filtered lines are only 1 μ W. The authors might wish to comment on the power efficiency of the approach.

We thank the reviewer for raising this important point regarding the power efficiency of the soliton microcomb source. In the present experiment, the total comb output power is on the order of 1 mW for a pump power of 1 W, corresponding to an overall pump-to-comb conversion efficiency of approximately 0.1%. This efficiency is consistent with typical single-ring SiN soliton microcombs optimized primarily for stable soliton formation rather than power extraction. Importantly, recent studies have demonstrated that the power efficiency of soliton microcombs can be dramatically improved without requiring ultra-high-Q resonators, by employing coupled-resonator architectures. In particular, conversion efficiencies as high as 30-50% have been achieved using coupled-ring designs with moderate Q factors on the order of 10^6 (Nature Photon. **17**, 992, 2023; APL Photonics **10**, 126119, 2025). These results

indicate that the relatively low conversion efficiency observed in the present work is not a fundamental limitation of soliton microcomb, but rather a design choice prioritizing robustness and stability. We therefore emphasize that integrating such high-efficiency coupled-resonator schemes represents a promising and realistic pathway toward substantially increasing the available optical power per comb line in future implementations. We have added a brief comment and appropriate references to the revised manuscript to clarify this point and place our results in the context of recent progress in high-efficiency microcomb generation. See Line 148-155 and Ref. 30,31.

2. Please clarify the IF frequency. The operating frequency range of the SHM is DC–40 GHz and the bandwidth of the real-time scope is also 40 GHz, whereas the demonstrated 42 Gbaud QPSK link already exceeds this bandwidth limit. It would be helpful to understand how the IF signal performs beyond 40 GHz. Additional details on the received signals would therefore be helpful, possibly including the spectrum of the received IF signal.

We thank the reviewer for this important and insightful comment regarding the IF bandwidth. In the present system, the sub-harmonic mixer (SHM) and the real-time oscilloscope both have a nominal IF bandwidth of DC-40 GHz. While the demonstrated QPSK symbol rate reaches 42 GBaud, this does not imply that frequency components beyond 40 GHz are directly processed. Instead, the overall IF chain, including the SHM, IF amplifiers, and oscilloscope frontend, effectively acts as a low-pass filter with a gradual roll-off near 40 GHz. As a result, higher-frequency spectral components of the IF signal are naturally attenuated, leading to bandwidth-limited signal reception. The impact of this effective low-pass filtering manifests as inter-symbol interference and amplitude/phase distortion, which are fully captured in the measured EVM. Importantly, even under this bandwidth-limited condition, the received QPSK signal at 42 GBaud remains below the hard-decision FEC threshold, demonstrating that the link performance is robust against the IF bandwidth constraint. We note that no signal components beyond the nominal IF bandwidth are explicitly relied upon in the performance evaluation. Rather, the reported results represent the achievable performance of the complete, bandwidth-limited receiver chain. To clarify this point, we have added additional explanation in the revised manuscript. See Line 214-217.

3. The offline post processing of the IF signal captured on the real-time oscilloscope should be described, including whether any equalization and frequency and phase tracking is employed.

We thank the reviewer for this important comment regarding signal processing. In the present experiment, no offline post processing was performed on the IF signals captured by the real-time oscilloscope. All signal processing was carried out using the built-in, real-time digital signal processing functions of the oscilloscope. Specifically, the oscilloscope performed real-time demodulation and EVM calculation based on its

internal processing chain. No additional offline equalization, frequency offset compensation, or phase tracking was applied beyond this real-time processing. The EVM values reported in the manuscript therefore correspond directly to the performance evaluated in real time by the receiver system, without any subsequent digital post-compensation. This approach ensures that the reported results faithfully reflect the intrinsic performance of the photonic THz link under practical receiver conditions. We added the sentence on this point. See Line 200-203.

4. It would be helpful if the BER performance could be measured, as this is a very basic and widely accepted performance metric. In addition, a comparison between the EVM and BER results would further strengthen the reliability of the presented performance evaluation. The duration of the samples taken using the real-time oscilloscope should also be stated, as this determines the minimum BER that can be inferred.

We thank the reviewer for raising this important point regarding BER evaluation. In the present experiment, direct BER measurement based on error counting was not feasible for technical reasons. Specifically, the acquisition length of the real-time oscilloscope is insufficient to collect the large number of bits required to reliably estimate low BER values. In addition, the built-in signal analysis of the oscilloscope is optimized for real-time EVM evaluation and does not provide access to reference-sequence synchronization or explicit error counting functionality. The present system is therefore designed and optimized for real-time EVM-based performance assessment rather than offline BER counting.

We emphasize that EVM is a widely accepted performance metric for high-speed coherent wireless links and is directly linked to hard-decision FEC thresholds, which are the relevant criteria for practical system operation. All reported transmission results are evaluated under bandwidth-limited, real-time receiver conditions without any offline digital post-compensation, ensuring a conservative and realistic assessment of link performance. Direct BER measurement over sufficiently long data sequences would certainly be valuable and is an important topic for future work.

5. Please clarify what contributes to the additive noise for the QPSK transmission performance. More detailed noise analysis would be beneficial.

We thank the reviewer for this insightful comment regarding the origin of the additive noise observed in the QPSK transmission. In the present system, the dominant contribution to additive noise at higher symbol rates arises from a gradual reduction in the effective signal-to-noise ratio as the occupied IF bandwidth increases. Specifically, the overall receiver chain, including the sub-harmonic mixer, IF amplifiers, and oscilloscope frontend, exhibits a finite and non-flat frequency response with a gradual roll-off near 40 GHz. As the symbol rate increases, this bandwidth limitation leads to

increased thermal and amplifier noise integration, as well as inter-symbol interference caused by amplitude and group-delay distortion. These effects manifest as isotropic constellation broadening and are therefore interpreted as additive noise in the EVM analysis. See Line 214-217.

Furthermore, the choice of symbol rates needs further explanation: QPSK is evaluated at 10, 20, 30, 40, and 42 Gbaud, while 16-QAM is tested at 5, 10, 15, 25, and 28 Gbaud. Using identical symbol rates across different modulation formats would enable direct performance comparison and cross-validation. For example, the manuscript states that phase noise dominates at 10 Gbaud and additive noise begins to dominate from 20 Gbaud in the QPSK link. Observing similar trends in the 16-QAM case would significantly strengthen the analysis.

We thank the reviewer for this thoughtful comment regarding the choice of symbol rates and cross-modulation comparison. In the present study, the symbol rates for each modulation format were selected to uniformly sample the practically accessible transmission window for that format. Specifically, for both QPSK and 16QAM, the symbol-rate ranges were first determined by the onset of bandwidth limitation and the hard-decision FEC threshold, and the ranges were then approximately divided into five representative operating points. This approach was chosen to efficiently capture the performance evolution and dominant impairment mechanisms across the full usable symbol-rate range, rather than to emphasize point-by-point comparison at identical rates.

We note that using identical symbol rates for different modulation formats can be potentially misleading, because QPSK and 16QAM have fundamentally different SNR and EVM requirements. At a given symbol rate, one format may operate well within the error-free regime while the other already exceeds the FEC threshold, which could obscure the underlying performance trends.

Importantly, consistent trends are observed across both modulation formats. In both QPSK and 16QAM, constellation distortion at lower symbol rates is dominated by carrier phase noise, whereas at higher symbol rates isotropic broadening emerges as additive noise becomes dominant due to bandwidth limitation and noise integration in the IF chain. In the case of 16QAM, this transition occurs at lower symbol rates owing to its higher sensitivity to amplitude noise. We have clarified the rationale for the choice of symbol rates in the revised manuscript. See Lines 211-214.

6. A more detailed comparison with the studies reported in references [25] and [26] would be valuable. The main difference appears to be the compact and robust SiN microcomb, which improves system stability. The 560 GHz carrier power in this work is 3.2 μ W, which is similar with the 3 μ W reported in [26], and the transmission path length is the same (10 mm). However, the data rate in [26] is limited to 1 Gbaud for QPSK with an EVM of 23.6%. It would therefore

be helpful if the authors could clarify why a significantly higher data rate is achievable in the present work, and what fundamental system-level improvements contribute to this performance enhancement.

We thank the reviewer for this important comment and for pointing out the need for a clearer comparison with Refs. [25] and [26]. Although the carrier frequency, transmitted THz power, and propagation distance are similar between Ref. [26] and the present work, the achievable data rate is fundamentally determined by the signal generation and detection architecture, which differs substantially between the two systems.

In Ref. [26], the THz signal is generated using double-sideband optical modulation and detected by square-law detection with a Schottky barrier diode (SBD). This approach inherently limits the effective modulation bandwidth and sensitivity, as the detected baseband signal relies on envelope detection without phase information. As a result, the system is constrained to low symbol rates (1 Gbaud QPSK), even though the radiated THz power is comparable.

In contrast, the present work employs in-phase and quadrature (IQ) modulation combined with sub-harmonic mixing (SHM), enabling true heterodyne detection with full access to both amplitude and phase information. This coherent architecture provides significantly higher receiver sensitivity, improved tolerance to noise, and a much wider effective bandwidth, which together support high-order modulation and multi-tens-of-GBaud operation. In addition, the electrical IF chain and receiver frontend were carefully optimized to support broadband coherent processing, further contributing to the substantial increase in achievable symbol rate.

We emphasize that the performance enhancement reported here does not arise from increased THz output power or propagation distance, but from a fundamental shift from envelope-based detection to coherent heterodyne transceiver operation, enabled by IQ modulation and SHM. We have clarified this system-level comparison and the key reasons for the higher achievable data rate relative to Refs. [25] and [26] in the revised manuscript. See Lines 110-112 and 245-247.

7. In the Discussion, it is stated that “frequency-agile transmission” has been achieved. The authors should explain and justify this statement. Isn’t the frequency fixed by the FSR of the micro-ring resonator? The paper does not demonstrate that this can be adjusted.

We thank the reviewer for pointing out this important issue. We agree that, in the present manuscript, frequency agility is not experimentally demonstrated.

In the original text, the term “frequency-agile transmission” was intended to convey the principle-level flexibility of photomixing-based THz generation using microcombs, whereby the carrier frequency can, in principle, be reconfigured by selecting different

pairs of comb lines for photomixing or by employing microresonators with different FSRs. However, as correctly noted by the reviewer, the experimental results presented in this work are restricted to a fixed carrier frequency determined by the 560 GHz FSR of the employed micro-ring resonator, and no active tuning or switching of the carrier frequency is demonstrated.

To avoid potential misunderstanding or overstatement, we have therefore modified the phrase “frequency-agile transmission” from the revised manuscript and clarified the discussion to focus strictly on the experimentally demonstrated performance at a fixed carrier frequency. We believe this revision improves the accuracy and clarity of the manuscript. See Line 245-247.

8. Extended Data Figure 3(a) shows a significant improvement in phase noise with the DFB lasers locked to the microcomb lines. However, the EVM improvement shown in constellation diagrams in the rest of that figure is relatively modest. Can the authors comment further on this? Can the residual phase noise of the optical injection locked system be quantified?

We thank the reviewer for this insightful comment regarding the apparent discrepancy between the strong phase-noise suppression observed in the RF spectra and the more modest improvement seen in the constellation diagrams. The pronounced phase-noise reduction in Extended Data Fig. 3(a) reflects the intrinsic narrowing of the THz carrier linewidth achieved by optical injection locking (OIL) to the soliton microcomb. In contrast, the constellation diagrams represent end-to-end link performance after reception and signal processing. In the present experiment, all signal processing was carried out using the built-in digital signal processing functions of the real-time oscilloscope, including carrier recovery and phase tracking. As a result, a significant fraction of low-frequency phase fluctuations is effectively compensated in real time, which partially masks the improvement in raw carrier phase noise at the constellation level. Consequently, the EVM improvement appears more modest than the phase-noise suppression observed in the RF spectral domain. The residual phase-noise characteristics of microcomb-based optical injection locking have been analyzed in detail in our previous work (Ref. [23]). We have clarified these points in the revised manuscript. See Line 266-268 and 279-283.

9. Other minor points: (a) In the Introduction, it is stated that free-space path loss scales with frequency. Significantly, and to be precise, it scales with the square of frequency, as the authors are no doubt aware.

We thank the reviewer for this helpful clarification. We agree that free-space path loss scales with the square of frequency, rather than linearly with frequency. Following the reviewer’s suggestion, we have revised the corresponding sentence in the Introduction to explicitly reflect the quadratic frequency dependence. See Line 55.

(b) It is stated that the spectral efficiency with QPSK is 4 bit/s/Hz and with 16QAM it is 8 bit/s/Hz. Considering the optical or wireless passband bandwidth, these values should be 2 bit/s/Hz and 4 bits/s/Hz respectively.

We thank the reviewer for pointing out this error. We agree that, when defined with respect to the occupied passband bandwidth, the spectral efficiencies should be 2 bits $\text{s}^{-1} \text{Hz}^{-1}$ for QPSK and 4 bits $\text{s}^{-1} \text{Hz}^{-1}$ for 16QAM. We have corrected the manuscript accordingly. See Lines 230 and 239.

(c) Please add reference numbers to Extended Data Table 1 so that readers can more easily locate the corresponding sources.

We thank the reviewer for this helpful suggestion. In accordance with the reviewer's comment, we have added reference numbers to Extended Data Table 1 so that the corresponding sources can be easily identified. See Extended Data Table 1 and Refs. 34-37.

Response to Reviewer 2's comment

The manuscript presents an experimental demonstration of high-speed wireless transmission at 560 GHz, employing a fiber-packaged silicon-nitride soliton microcomb as a compact, low-phase-noise optical frequency reference. The authors achieve a single-channel data throughput of up to 112 Gbps using QPSK and 16QAM modulation, establishing a record transmission rate in the 560 GHz band. The work addresses important challenges associated with next-generation THz wireless backhaul for 6G networks and demonstrates a tangible proof-of-concept for realizing >100 Gbps-class wireless links above 350 GHz. The paper is in good structure, and the English is well written.

However, besides the novelty and results, the manuscript requires major revisions for clarity, to ensure the impact and reproducibility of the work. To guide the authors in this process, I list several more specific comments to be addressed:

We thank the reviewer for the positive assessment of the significance and potential impact of our work. We have carefully addressed all comments raised below and revised the manuscript accordingly to improve clarity, completeness, and reproducibility. Detailed responses to each point are provided below. All modifications are highlighted in yellow in the revised manuscript for ease of review.

1) On page 8, line 148, the authors claimed the UV-bonded joint is robust up to 3 W, could the authors also compare if the coupling loss got changed comparing from 1 W to 3 W?

We thank the reviewer for this important question regarding coupling stability at high pump powers. As described in the Methods section under “High-power endurance test of the UV-bonded joint,” the high-power test was intentionally performed without the microresonator chip in order to isolate the fiber-waveguide bonding interface from resonator-related thermal or nonlinear effects. In this configuration, the fractional optical power change directly reflects variations in the fiber-to-waveguide coupling loss. As shown in the endurance test, no measurable increase in fractional loss was observed when the launched power was increased from 1 W to 3 W, indicating that the coupling loss remains unchanged within the experimental resolution. We have clarified in the main text that the reported fractional loss change in the endurance test corresponds directly to the coupling-loss stability of the UV-bonded interface under high-power operation. See Line 389-390.

2) On page 8, line 151, the authors pointed out that the coupling efficiency of the fiber-coupled system is 38.8%, however, it cannot be extracted from Fig. 2b.

We thank the reviewer for pointing out this important source of ambiguity. We agree that, in the original version, Fig. 2b showed the normalized temporal variation of the output power referenced to its initial value, and therefore did not directly represent the absolute coupling efficiency. The simultaneous use of normalized output power and

coupling efficiency indeed had the potential to cause confusion. To address this issue, we have revised Fig. 2b to explicitly present the time evolution of the coupling efficiency, rather than normalized output power. Correspondingly, the related text has been revised to clearly distinguish between absolute coupling efficiency and its temporal stability. With this revision, the reported value of 38.8% coupling efficiency is now directly supported by the figure and its caption. See **Lines 162-167, 401-402, and 649-653** as well as Fig. 2b.

3) On page 8, line 153, can the authors explain where the $\pm 2.54\%$ fluctuation comes from and why it looks like it follows a certain period?

We thank the reviewer for this insightful question regarding the origin and periodicity of the $\pm 1.27\%$ ($\pm 2.54\%$ in the previous manuscript) fluctuation observed in the free-space lensed-fibre coupling. This fluctuation originates from mechanical instability inherent to the free-space coupling configuration. In particular, we attribute the quasi-periodic variation (with a characteristic period of approximately 10 minutes) to weak mechanical vibrations transmitted through the optical table. During the measurement, other experimental equipment placed on the same optical table—including systems equipped with compressors—were operating intermittently, and their vibration is believed to induce periodic beam-pointing fluctuations at the lensed fibre tip. Such beam-pointing instability directly translates into coupling-efficiency variation in free-space coupling schemes. In contrast, the direct fibre-coupled interface is mechanically fixed and therefore immune to these environmental perturbations, resulting in the substantially improved stability observed in Fig. 2b. We have clarified this point in the revised manuscript. See **Line 167-171**.

4) On page 8, line 165, the authors mentioned Fig. 2d is indistinguishable, while it seems it is clearly not the case.

We thank the reviewer for pointing out this ambiguity. We agree that the term “indistinguishable” was too strong and could be misleading. Our intention was not to claim that the spectra are identical in every detail, but rather that both configurations generate soliton combs with broadly similar spectral characteristics. Importantly, despite their comparable spectral profiles, the two coupling schemes exhibit a pronounced difference in soliton lifetime. To avoid overstatement and improve clarity, we have revised the text to replace “indistinguishable spectrum” with “spectrally similar soliton comb” and clarified that the key contrast lies in the markedly different operational stability. See **Line 180-182**.

5) On page 10, line 208, the authors wrote the data at 28 GBaud has EVM of 12.5%, whereas it seems above the HD-FEC limit of 12.9%, please double-check.

We thank the reviewer for pointing out this inconsistency. We rechecked the 28-GBaud 16QAM dataset and found that the EVM value reported in the text (12.5%) was incorrect. The correct EVM at 28 GBaud is 12.9%, which is consistent with the HD-FEC limit (12.9%). The 28-GBaud result therefore lies at the threshold and still satisfies the HD-FEC criterion. We have corrected the text accordingly. See Line 237.

6) On page 11, line 232, the authors wrote “The use of microcomb-based OIL significantly improves the phase coherence of photomixed THz carriers.” Did the authors also perform the phase noise measurement or extracted the phase noise besides the comparison of the RF spectrum in Extended Data Figure 3a?

We thank the reviewer for this insightful comment regarding the phase noise of the THz carrier. In the present work, the improvement in phase coherence is experimentally confirmed through RF spectral measurements, as shown in Extended Data Fig. 3a, which directly reflect the linewidth narrowing of the photomixed THz carrier enabled by microcomb-based optical injection locking. A full quantitative phase-noise characterization (e.g., phase-noise power spectral density versus offset frequency) was not newly performed in this study. Instead, the detailed phase-noise characteristics of microcomb-based optical injection locking—including residual phase noise and transfer properties—have been thoroughly analyzed and reported in our previous work (Ref. [23]). To avoid ambiguity, we have clarified in the revised manuscript that the present paper relies on RF spectral evidence and prior quantitative phase-noise analysis reported in Ref. [23]. See Line 266-268 and Ref. 23.

7) On page 14, line 296, the authors mentioned they optimized the MFD matching, could authors provide a measured coupling loss value?

We thank the reviewer for this helpful request. The coupling loss associated with the optimized MFD matching was experimentally measured to be less than 0.3 dB for the bonded fibre–fibre interface. We have now explicitly stated this measured coupling loss in the revised manuscript. See Line 334.

8) On page 15, line 347, 351, why the coupling efficiency of the high-NA fiber coupled system (38.8%) is worse than the lensed-fiber coupled system (50%)?

We thank the reviewer for this insightful question regarding the difference in coupling efficiency between the high-NA fiber-coupled and lensed-fiber-coupled configurations. The difference originates from the degree of mode-field matching between the incident optical field and the effective aperture of the access waveguide of the microresonator. In the lensed-fiber configuration, the output beam size and wavefront can be finely adjusted by the lens geometry and alignment, allowing closer matching to the effective mode size of the inverse-tapered access waveguide. As a result, a higher initial

coupling efficiency of approximately 50% can be achieved. In contrast, in the high-NA fiber direct-connection scheme, the coupling efficiency is primarily determined by the intrinsic MFD of the high-NA SMF and its overlap with the effective input aperture of the access waveguide. Several factors contribute to the somewhat lower coupling efficiency of 38.8%, including (i) residual MFD mismatch between the high-NA SMF and the waveguide mode, (ii) diffraction effects at the inverse taper, (iii) beam expansion within the adhesive layer, and (iv) small transverse or angular misalignments introduced during permanent bonding.

Importantly, while the lensed-fiber approach enables higher initial coupling efficiency through active alignment, it is highly sensitive to thermal drift and mechanical vibration, leading to coupling degradation over time. By contrast, the high-NA fiber-coupled interface sacrifices some initial coupling efficiency in exchange for exceptional mechanical robustness and long-term stability, which are critical for sustained high-power pumping and continuous soliton microcomb operation. We have clarified this physical origin of the coupling-efficiency difference in the revised manuscript. See **Line 406-410**.

And since the output power of the lensed-fiber coupled system is higher $P_{out}=250\text{ mW}$, why in Fig. 2b the blue curve has lower value?

9) On page 28, line 585 to 588, the coupling efficiency of 38.8% is not clear to be seen on Fig. 2b. Same for the lensed-fiber case, where the authors mentioned the degradation from 50% to 43%.

We thank the reviewer for pointing out this potential source of confusion. In the original version of the manuscript, Fig. 2b did not show the absolute output power or coupling efficiency. Instead, it plotted the output power normalized to its initial value, in order to emphasize temporal stability rather than absolute coupling efficiency. Consequently, both coupling schemes were normalized to 100% at the start of the measurement, and the blue curve appearing at a lower level does not indicate a lower absolute output power, but rather a larger relative degradation over time.

We agree that presenting normalized output power alongside absolute coupling-efficiency values can be misleading. To eliminate this ambiguity, we have revised Fig. 2b to explicitly display the time evolution of the coupling efficiency, and we have revised the corresponding text to clearly distinguish between absolute coupling efficiency and its temporal fluctuation. These changes ensure that the higher initial output power of the lensed-fiber system and its larger temporal instability are correctly interpreted. All related modifications are highlighted in the revised manuscript. See **Lines 162-167, 401-402, and 649-653** as well as Fig. 2b.

10) On page 29, line 594, the authors mentioned “degraded coupling”, which is confusing.

since it seems the free-space lensed-fiber coupling loss is lower than the high-NA fiber-coupled system

We thank the reviewer for pointing out this ambiguity. We agree that the phrase “degraded coupling” was inappropriate and potentially misleading in this context, because the free-space lensed-fiber configuration indeed exhibits a higher initial coupling efficiency than the high-NA fiber-coupled system. Our intention was to describe the temporal degradation of coupling stability rather than a lower absolute coupling efficiency. To avoid confusion, we have removed the phrase “degraded coupling” from the revised manuscript and rephrased the text to clearly distinguish between initial coupling efficiency and long-term coupling stability. See **Line 660**.

11) In general, based on points 1), 2), 7), 8), 9), 10), I suggest the authors compare the coupling loss of the presented two systems, and show a clear loss breakdown to prove the results. The high-NA fiber-coupled system seems more stable, but the coupling loss is higher, can authors further explain?

We thank the reviewer for this important suggestion regarding a quantitative comparison of coupling loss between the two coupling schemes. In response, we have carried out a detailed loss breakdown for both the direct high-NA fiber-coupled system and the lensed-fiber free-space coupling system, which is now clarified in the revised manuscript.

For the direct high-NA fiber-coupled system, an input power of 480 mW and an output power of 186 mW correspond to a coupling efficiency of 38.8%, yielding a final insertion loss of approximately 4.1 dB. The FC/APC connector pair contributes approximately 0.2–0.3 dB, and the standard SMF-to-high-NA-SMF fusion splices contribute less than 0.3 dB each, while propagation losses in the fibers and on-chip waveguides are negligible. Taking these contributions into account, the remaining loss is attributed to the fiber-to-chip interfaces, giving an estimated facet loss of approximately 1.6 dB per facet (input and output combined).

For the lensed-fiber free-space coupling system, an input power of 480 mW and an output power of 240 mW correspond to a coupling efficiency of 50%, yielding a final insertion loss of approximately 3.0 dB. After accounting for the FC/APC connector loss of approximately 0.2–0.3 dB and negligible propagation losses, the estimated fiber-to-chip facet loss is approximately 1.4 dB per facet.

These results show that the lensed-fiber configuration achieves a slightly lower initial coupling loss due to the ability to actively optimize beam size and alignment. In contrast, the direct high-NA fiber-coupled system exhibits a modestly higher insertion loss, primarily originating from fixed mode-field mismatch, diffraction at the inverse taper, beam expansion in the adhesive layer, and small residual transverse or angular misalignment introduced during permanent bonding. Importantly, this small penalty in

coupling loss is compensated by a substantial improvement in mechanical robustness and long-term stability, as demonstrated by the significantly reduced coupling fluctuations over extended operation. We therefore emphasize that the two approaches represent a trade-off between initial coupling efficiency and long-term stability, and that the high-NA fiber-coupled interface is particularly advantageous for high-power, long-duration soliton microcomb operation. We added the sentence on this point. See **Line 345-361**.

12) On page 33, Figure 6, can the authors include the point of the experimental data in the simulated plot, since the simulation is based on the experimental setting (e.g. page 18, line 395, “output power of 3.2 μW ”). Does the experiment data match with the simulation?

We thank the reviewer for this valuable suggestion. In response, we have added the experimentally measured data point (EVM = 11.6% at a propagation distance of 10 mm) to the simulated plot in Fig. 6. While the simulation predicts an EVM of approximately 12% at a longer distance of 30 mm under the same transmitted THz power, the purpose of the simulation is not to exactly reproduce the experimental condition, but rather to explore the achievable transmission reach under idealized and controlled assumptions based on the experimental parameters.

The fact that the experimentally measured EVM at 10 mm lies within the same EVM range as the simulated results demonstrates good qualitative agreement and confirms that the simulation provides a reasonable and conservative projection of distance scaling. We have clarified this point in the revised manuscript and updated Fig. 6 accordingly. See **Lines 300-305 and 701-703** as well as **Fig. 6**.

13) For all the constellation diagrams (page 31 Figure 4, page 32 Figure 5, page 36 Extended Data Figure 3), can the authors indicate clearly the symbol rate? Especially for page 36 Extended Data Figure 3, please indicate clearly the comparison between the OIL-driven and free-running cases.

We thank the reviewer for this helpful suggestion. In response, we have revised all constellation diagrams in Fig. 4, Fig. 5, and Extended Data Fig. 3 to clearly indicate the corresponding symbol rates. These revisions improve clarity and enable direct visual comparison of transmission performance. See **Fig. 4, Fig. 5, and Extended Data Fig. 3**.

14) On page 30, Figure 3. Can the authors improve the schematic with conventional symbols, and also include the picture of the setup at the Transmitter and Receiver side if possible?

We thank the reviewer for this helpful suggestion regarding Fig. 3. In response, we have revised Fig. 3 using more conventional and standardized symbols for photonic and RF components, improving clarity and consistency with common schematic

representations. In addition, representative photographs of the experimental setup at the transmitter and receiver sides have been added as insets to the revised figure. These changes enhance the readability of the system architecture without overloading the main schematic. See **Fig. 3**.

The manuscript presents an experimental demonstration of high-speed wireless transmission at 560 GHz, employing a fiber-packaged silicon-nitride soliton microcomb as a compact, low-phase-noise optical frequency reference. The authors achieve a single-channel data throughput of up to 112 Gbps using QPSK and 16QAM modulation, establishing a record transmission rate in the 560 GHz band. The work addresses important challenges associated with next-generation THz wireless backhaul for 6G networks and demonstrates a tangible proof-of-concept for realizing >100 Gbps-class wireless links above 350 GHz. The paper is in good structure, and the English is well written.

However, besides the novelty and results, the manuscript requires major revisions for clarity, to ensure the impact and reproducibility of the work. To guide the authors in this process, I list several more specific comments to be addressed:

- 1) On page 8, line 148, the authors claimed the UV-bonded joint is robust up to 3 W, could the authors also compare if the coupling loss got changed comparing from 1 W to 3 W?
- 2) On page 8, line 151, the authors pointed out that the coupling efficiency of the fiber-coupled system is 38.8%, however, it cannot be extracted from Fig. 2b.
- 3) On page 8, line 153, can the authors explain where the $\pm 2.54\%$ fluctuation comes from and why it looks like it follows a certain period?
- 4) On page 8, line 165, the authors mentioned Fig. 2d is indistinguishable, while it seems it is clearly not the case.
- 5) On page 10, line 208, the authors wrote the data at 28 GBaud has EVM of 12.5%, whereas it seems above the HD-FEC limit of 12.9%, please double-check.
- 6) On page 11, line 232, the authors wrote "The use of microcomb-based OIL significantly improves the phase coherence of photomixed THz carriers." Did the authors also perform the phase noise measurement or extracted the phase noise besides the comparison of the RF spectrum in Extended Data Figure 3a?
- 7) On page 14, line 296, the authors mentioned they optimized the MFD matching, could authors provide a measured coupling loss value?
- 8) On page 15, line 347, 351, why the coupling efficiency of the high-NA fiber coupled system (38.8%) is worse than the lensed-fiber coupled system (50%)? And since the output power of the lensed-fiber coupled system is higher $P_{out}=250$ mW, why in Fig. 2b the blue curve has lower value?
- 9) On page 28, line 585 to 588, the coupling efficiency of 38.8% is not clear to be seen on Fig. 2b. Same for the lensed-fiber case, where the authors mentioned the degradation from 50% to 43%.
- 10) On page 29, line 594, the authors mentioned "degraded coupling", which is confusing, since it seems the free-space lensed-fiber coupling loss is lower than the high-NA fiber-coupled system

- 11) In general, based on points 1), 2), 7), 8), 9), 10), I suggest the authors compare the coupling loss of the presented two systems, and show a clear loss breakdown to prove the results. The high-NA fiber-coupled system seems more stable, but the coupling loss is higher, can authors further explain?
- 12) On page 33, Figure 6, can the authors include the point of the experimental data in the simulated plot, since the simulation is based on the experimental setting (e.g. page 18, line 395, "output power of 3.2 μW "). Does the experiment data match with the simulation?
- 13) For all the constellation diagrams (page 31 Figure 4, page 32 Figure 5, page 36 Extended Data Figure 3), can the authors indicate clearly the symbol rate? Especially for page 36 Extended Data Figure 3, please indicate clearly the comparison between the OIL-driven and free-running cases.
- 14) On page 30, Figure 3. Can the authors improve the schematic with conventional symbols, and also include the picture of the setup at the Transmitter and Receiver side if possible?